Temporal and spatial distribution of histone acetylation in mouse molar development

Du Wen 1
Luo Wanyi 2
Zheng Liwei 3
Zhou Xuedong 2
Du Wei 2 weidu@scu.edu.cn
1 State Key Laboratory of Oral Diseases & National Center for Stomatology & National Clinical Research Center for Oral Diseases, Department of Prosthodontics II, West China Hospital of Stomatology, Sichuan University , Chengdu, Sichuan , China
2 State Key Laboratory of Oral Diseases & National Center for Stomatology & National Clinical Research Center for Oral Diseases, Department of Cariology and Endodontics, West China Hospital of Stomatology, Sichuan University , Chengdu, Sichuan , China
3 State Key Laboratory of Oral Diseases & National Center for Stomatology & National Clinical Research Center for Oral Diseases, Department of Pediatric Dentistry, West China Hospital of Stomatology, Sichuan University , Chengdu, Sichuan , China
Reno Philip
Electronic publication date: 2025 Mar 31
Publication date: 2025
Volume: 13
Electronic Location ID: e19215
Received 2024 Nov 4; Accepted 2025 Mar 5
Copyright: © 2025 Du et al.
Copyright year: 2025
Copyright holder: Du et al.
License: This is an open access article distributed under the terms of the Creative Commons Attribution License, which permits unrestricted use, distribution, reproduction and adaptation in any medium and for any purpose provided that it is properly attributed. For attribution, the original author(s), title, publication source (PeerJ) and either DOI or URL of the article must be cited.
License URL: https://creativecommons.org/licenses/by/4.0/

Keywords: Histone acetylation, Epigenetic modification, Tooth, Molar development

Funding: National Natural Science Foundation of China 82201003 Sichuan Science and Technology Program 2025ZNSFSC0768 Research Funding from West China Hospital of Stomatology Sichuan University RCDWJS2023-11 This work was supported by the National Natural Science Foundation of China (No. 82201003), Sichuan Science and Technology Program (No. 2025ZNSFSC0768), and the Research Funding from West China Hospital of Stomatology Sichuan University (No. RCDWJS2023-11). The funders had no role in study design, data collection and analysis, decision to publish, or preparation of the manuscript.

==============================
Histone acetylation is one of the most widely studied histone modification, regulating a variety of biological activities like organ development and tumorigenesis. However, the role of histone acetylation in tooth development is poorly understood. Using the mouse molar as a model, we mapped the distribution patterns of histone H3 and H4, as well as their corresponding acetylation sites during tooth formation in order to unveil the connection between histone acetylation modification and tooth development. Moreover, key histone acetyltransferases and histone deacetylases were detected in both epithelial and mesenchymal cells during tooth development by scRNA-seq and immunohistochemistry. These results suggest that histone acetylation modification functions as an important mechanism in tooth development at different stages.

Introduction

Tooth development is a complex and continuous process that results from the interaction between epithelial cells derived from the ectoderm, and mesenchymal cells derived from the cranial neural crest. This process begins with the phrase of tooth germ and then undergoes the formation of dental lamina, bud stage, cap stage and bell stage, with different tooth tissues and shapes formed gradually (Yu & Klein, 2020). Multiple factors (genetic, epigenetic and environmental factors) are known to precisely regulate this process (Jaenisch & Bird, 2003; Jussila & Thesleff, 2012; Zhong, Tian & Gao, 2021). As the regulation of signaling molecules in tooth development has been relatively well studied, the gene expression patterns and mechanisms of epigenetic regulation in tooth development have attracted more attention.

Epigenetic modification, an important regulatory method in directional cell differentiation and specific gene expression, consists of histone modification, DNA methylation, RNA modification, and non-coding RNA (Zhong, Tian & Gao, 2021). Histone modification, the post-transcriptional modification including acetylation, methylation, phosphorylation, and ubiquitination, has been extensively studied (Strahl & Allis, 2000; Jenuwein & Allis, 2001; Kouzarides, 2007). Among them, histone acetylation is one of the most important mechanisms for gene transcription regulation that mostly occurs on lysine residues within the N-terminal tail protruding from core histone. It is mainly catalyzed by histone acetyltransferase (HAT) and histone deacetylase (HDAC) (Pogo, Allfrey & Mirsky, 1966; Clayton et al., 1993; Yang & Seto, 2008; Marmorstein & Zhou, 2014). While the majority of studies on histone acetylation have focused on odontogenic differentiation (Wang et al., 2014; Tao et al., 2019a; Yamauchi, Shimizu & Duncan, 2024), which occurs during the late stage of tooth development, relatively limited research has been conducted on histone acetylation during the earlier stages of tooth development. The expression of a gene or epigenetic mark during the development of a specific organ is generally considered to suggest a potential biological role in that process. Therefore, in the current study, mouse molar is used as a model and we detect the distribution patterns of histone H3 and H4, as well as their corresponding acetylation sites, at different developmental stages, in order to unveil the connection between histone acetylation modification and tooth formation. Moreover, by analyzing a published scRNA-seq dataset and immunohistochemistry, we found key HATs and HDACs were broadly expressed in both epithelial and mesenchymal cells during tooth development. Together our results provide a basis and new understanding for epigenetic modification in tooth development and regeneration.

Materials and Methods

Mouse lines and procedures

C57BL/6 mice were provided from the Experiment Animal Center of Sichuan University, and used at various embryonic and early postnatal stages. All mice were housed in specific pathogen-free facilities at the Experimental Animal Core of West China Hospital Sichuan University, a temperature-controlled (25 °C) environment with a 12-h light/dark cycle, cotton batting, and free access to food and water as previously described (Yu et al., 2022). Mice were mated overnight to generate embryos for experiments. Vaginal plug discovery was taken as E0.5 at noon. Pregnant mice were euthanized by CO2 followed by cervical dislocation at desired timepoints, and embryos were removed from the uterus. Both male and female embryos or postnatal pups from 3–4 different litters each timepoint were selected at random. Each experiment was performed at least three times with different embryos. The developmental stages of the tooth germs were judged from the tissue sections according to morphological criteria. All mice received humane treatment, and every possible measure was taken to reduce any potential distress. All experiments involving mice were approved by the Ethics Review Committee of the West China School of Stomatology, Sichuan University (WCHSIRB-D-2022-155).

Tissue preparation for sectioning

Tissue was prepared for paraffin sections by fixing embryonic heads or postnatal jaws in 4% paraformaldehyde (PFA) in PBS overnight at 4 °C and were then decalcifying in RNase-free EDTA for 3–5 days. Samples were embedded in paraffin and then sectioned at 6 µm.

Immunofluorescence and immunohistochemistry staining

Immunofluorescence staining is performed as previously described (Du et al., 2024). Briefly, paraffin sections were rehydrated through serial ethanol and water washes, and antigen retrieval was performed by incubation in pH 6.2 citric buffer containing 2 mM EDTA, 10 mM citric acid, 0.05% Tween 20 just below boiling temperature for 15 min followed by a 30-min cool-down to room temperature. Samples were blocked in 1X animal-free blocker (Vector Laboratories), supplemented with 2.5% heat inactivated goat serum, 0.02% SDS and 0.1% Triton-X for 1 h. Slides were then incubated with primary antibodies overnight at 4 °C. All the antibodies were diluted in the same blocking solution without serum. Primary antibodies and dilutions used are as follows: Histone H3 (acetyl K4+K9+K14+K18+K23+K27) (1:200; Abcam, ab300641), Histone H4 (acetyl K5 + K8 + K12 + K16) (1:200; Abcam, ab177790), Histone H3 (acetyl K27) (1:200; Abcam, ab4729), Histone H3 (acetyl K9) (1:200; Abcam, ab32129), Histone H4 (acetyl K5) (1:200; Abcam, ab51997), Histone H4 (acetyl K16) (1:200; Abcam, ab109463), CREBBP (1:100; Abcam, ab253202), HDAC1 (1:100; Abcam, ab109411), HDAC2 (1:100; Abcam, ab32117) and EP300 (1:100; Zenbio, 347220). Signals were detected using secondary antibodies conjugated with Alexa Fluor 555 (1:250; Thermo Scientific, Waltham, MA, USA) or horseradish peroxidase (HRP) followed by DAB staining kit (PK10006; Proteintech). Nuclear counterstaining was performed using DAPI (Thermo Scientific, Waltham, MA, USA) or hematoxylin, and mounted with antifade mountant (Solarbio) or neutral balsam. Images were taken using a confocal laser scanning microscope (FV3000; Olympus) or Slideview VS200 (Olympus).

scRNA-seq data analysis

scRNA-seq data was downloaded from the NCBI GEO database (GSE162413). Cell-level transcripts were processed using the Seurat package for normalization, quality control, dimensionality reduction, and clustering. The GSE162413 dataset comprises tooth germs collected at different developmental stages. Four stages were selected: bud stage (E12.5), cap stage (E14.5), bell stage (E16.5), and postnatal day 1 (PN1), including a total of 38,874 cells. The differentially expressed genes for each cluster, compared to other clusters at different time points, were identified by the FindAllMarkers function in Seurat, with a threshold of average log2FC > 0.25 and an adjusted p-value < 0.05 using two-sided Wilcoxon rank-sum test with a Bonferroni correction. Sub-clustering analyses were carried out using the Findneighbors function of the Seurat package with proper resolutions. Identified cell clusters and sub-clusters were visualized on UMAP plots, as previously described (Zeng et al., 2024). Clusters were annotated based on top expressed genes within each cluster and known canonical cell markers. After annotation, the expression of histone acetylation-related was further analyzed across epithelial, mesenchymal, and other cells types.

Image analysis

The average immunofluorescence pixel intensity of dental epithelium and mesenchyme or dental papilla (as defined in Fig. 1) were determined by measuring the total pixel intensity in the region of interest and divided by the total measured pixel area using the “Measure” function of ImageJ. The measured regional histone H3, histone H4, H3K9ac, H3K27ac, H4K5ac, H4K16ac, CREBBP, HDAC1, HDAC2 and EP300 levels in the epithelium were normalized across samples using averaged signals from the mesenchyme in the same section sample, respectively.

Figure 1 Schematic depiction of mouse molar development.

Statistical analysis

Data were collected as previously described (Du et al., 2024). Briefly, all experiments were repeated using three independent biological samples. Representative images were shown in figures (replicate staining images of independent biological samples are available in the author’s response letter in peer review reports from PeerJ website). Each data point in bar graphs represents a single biological sample without investigator blinding. No data were excluded. All statistical analyses were performed using the Prism 9 software and displayed as mean ± S.D (standard deviation) in graphs. All p values were calculated using unpaired two tailed Student’s t-test or one-way ANOVA followed by Dunnett’s test. Significance was taken as p < 0.05 with a confidence interval of 95%. *p < 0.05; **p < 0.01; ***p < 0.001; ****p < 0.0001.

Results

Spatial–temporal pattern of histone acetylation marks in mouse molar development

To investigate the distribution patterns of histone acetylation, we first performed immuno-fluorescence staining of histone 3 and 4 acetylation (H3Ac and H4Ac) in the mouse molar through the bud stage (E13.5), cap stage (E14.5), early bell stage (E16.5) and late bell stage (E18.5 and P3) (Figs. 1, 2, Figs. S1A, S2A). At E13.5, H3ac was weakly detected in both dental epithelium and mesenchyme, with few strong positive cells in the epithelium (Figs. 2A, 2F). In contrast, H4ac was ubiquitously expressed, pointing to a dominant role for H4ac in the early stage of molar development (Figs. 2K, 2P). At E14.5, both H3ac and H4ac were detected in the dental epithelium and the condense mesenchyme (Figs. 2B, 2G, 2L, 2Q). Instead of comparatively higher expression level in the condense mesenchyme at E14.5, H3ac localized in both dental epithelium and the dental papilla at similar expression level at E16.5 (Figs. 2C, 2H). While H4ac also localized in both dental epithelium and dental papilla at E16.5, it was weaker in the inner enamel epithelium (IEE) and the mesenchyme next to the IEE (Figs. 2M, 2R). However, the distribution of H3ac and H4ac were observed abundantly in the dental epithelium, but began to downregulate in the dental papilla at E18.5 (Figs. 2D, 2I, 2N, 2S). Of note, H3ac and H4ac distributions were apparently at a higher levels in the IEE, indicating the potential important role for participating amelogenesis. Similar distribution patterns of H3ac and H4ac were detected at later late bud stage, P3, with more apparent restricted in the IEE and ameloblasts. In the mesenchyme, H3ac and H4ac were mostly expressed in the odontoblasts (Figs. 2E, 2J, 2O, 2T).

Figure 2 The spatiotemporal pattern of histone 3 and 4 acetylation during tooth development.

(A–J ) Immunostaining of histone 3 acetylation (H3ac) on coronal sections of the developing molar germ (lingual to the left, buccal to the right) at E13.5 (A, F), E14.5 (B, G), E16.5 (C, H), E18.5 (D, I) and P3 (E, J). (K–T) Immunostaining of histone 4 acetylation (H4ac) on coronal sections of the developing molar germ at E13.5 (K, P), E14.5 (L, Q), E16.5 (M, R), E18.5 (N, S) and P3 (O, T). Dashed lines outline the molar epithelium. Representative images are shown. Scale bar in (T) represents 20 µm in (A, B, F, G, K, L, P, Q), 10 µm in (C, D, H, I, M, N, R, S), and 4 µm in (E, J, O, T).

We then set out to examine H3K9ac and H3K27ac as major histone 3 acetylation marks in mouse molar across different developmental stages (Fig. 3, Figs. S1B, S1C). At E13.5, weak signal of H3K9ac and H3K27ac was present in both dental epithelium and mesenchyme (Figs. 3A, 3F, 3K, 3P). At E14.5, H3K9ac was significantly upregulated in both the dental epithelium and mesenchyme (Figs. 3B, 3G). For H3K27ac, strong signal was restricted in the condense mesenchyme, with weak signal in the dental epithelium (Figs. 3L, 3Q). At E16.5, both H3K9ac and H3K27ac widely expressed in both the dental epithelium and mesenchyme. Specifically, H3K9ac in the IEE was much weaker than that in other cells, while H3K27ac was stronger in the IEE in the dental epithelium and dental papilla (Figs. 3C, 3H, 3M, 3R). At E18.5, H3K9ac was no longer highly expressed in the entire dental papilla, but rather in the entire dental epithelium including IEE and the mesenchymal cells underlying the IEE (Figs 3D, 3I). At this stage, H3K27ac was highly expressed in the IEE and the mesenchymal cells underlying the IEE, where weaker signal was also detected in other cells in the dental epithelium (Figs. 3N, 3S). At P3, H3K9ac was broadly expressed with higher expression in the IEE and the odontoblasts, and H3K27ac was ubiquitously expressed in both the dental epithelial and mesenchymal cells (Figs. 3E, 3J, 3O, 3T).

Figure 3 Disribution of H3K9ac and H3K27ac during tooth development.

(A–J) Immunostaining of H3K9ac in developing molar tooth germ at E13.5 (A, F), E14.5 (B, G), E16.5 (C, H), E18.5 (D, I) and P3 (E, J). (K–T) Immunostaining of H3K27ac in developing molar tooth germ at E13.5 (K, P), E14.5 (L, Q), E16.5 (M, R), E18.5 (N, S) and P3 (O, T). Dashed lines outline the molar epithelium. Representative images are shown. Scale bar in (T) represents 20 µm in (A, B, F, G, K, L, P, Q), 10 µm in (C, D, H, I, M, N, R, S), and 4 µm in (E, J, O, T).

We next examined the distribution of histone 4 acetylation marks H4K5ac and H4K16ac in mouse molar development (Fig. 4, Figs. S2B, S2C). At E13.5, both H4K5ac and H4K16ac were barely detectable (Figs. 4A, 4F, 4K, 4P). Increased signal of both H4K5ac and H4K16ac were obtained at E14.5. At this stage, while H4K5ac was relatively stronger in the condense mesenchyme than that in the epithelium (Figs. 4B, 4G), H4K16ac was present extensively expressed in both the dental epithelium and mesenchyme. Of note, the presence of H4K16ac was stronger in the outer enamel epithelium (OEE) and lower in the stellate reticulum (Figs. 4L, 4Q). At E16.5, H4K5ac and H4K16ac remain broadly distributed. Interestingly, slightly weak H4K5ac was observed in the OEE and dental papilla, where significant high H4K16ac was detected (Figs. 4C, 4H, 4M, 4R). At E18.5, the distribution of H4K5ac and H4K16ac in the dental papilla weakened, whereas that in the dental epithelium continued to be highly expressed. Moreover, both H4K5ac and H4K16ac were highly expressed in the IEE (Figs. 4D, 4I, 4N, 4S). At P3, H4K5ac and H4K16ac were still widely expressed in the dental epithelium cells, concentratedly in the IEE and tall-columnar ameloblasts, with the highest expression near the cervical loop (Figs. 4E, 4J, 4O, 4T). In the mesenchyme, H4K5ac was mainly expressed in the odontoblasts and in the middle zone between the two lateral cervical loops (Figs. 4E, 4J), while H4K16ac was widely expressed (Figs. 4O, 4T).

Figure 4 Distribution of H4K5ac and H4K16ac during tooth development.

(A–J) Immunostaining of H4K5ac in developing molar tooth germ at E13.5 (A, F), E14.5 (B, G), E16.5 (C, H), E18.5 (D, I) and P3 (E, J). (K–T) Immunostaining of H4K16ac in developing molar tooth germ at E13.5 (K, P), E14.5 (L, Q), E16.5 (M, R), E18.5 (N, S) and P3 (O, T). Dashed lines outline the molar epithelium. Representative images are shown. Scale bar in (T) represents 20 µm in (A, B, F, G, K, L, P, Q), 10 µm in (C, D, H, I, M, N, R, S), and 4 µm in (E, J, O, T).

Histone acetylation-related enzymes expressed in dental epithelium and mesenchyme during mouse molar development

Histone acetylation is accomplished by its modification enzymes, HATs and HDACs. To evaluate the specific alternations of HATs and/or HDACs, which medicate the acelylation of H3K9, H3K27, H4K5 and H4K16 (Fig. 5A) (Steunou, Rossetto & Côté, 2014; Fang et al., 2021), through the tooth development stages, we then utilized the published scRNA-seq data (Hu et al., 2022) for analysis. We found HATs and HDACs were widely present in both dental epithelium and mesenchyme during tooth development. Notably, at each timepoint, HATs and HDACs exhibited stronger signals in the mesenchyme, and Crebbp and Hdac2 were the prominent HATs and HDACs, respectively (Figs. 5B–5E). Rtt09, Hat1, Atac2, Sas2 and Esa1 were barely detectable in the scRNA-seq, suggesting that they are either expressed at very low levels or not at all. We next examined the distribution of CREBBP, EP300, HDAC1 and HDAC2, the enzymes with high expression levels in scRNA-seq, by immunohistochemistry (Fig. 6, Fig. S3). As expected, CREBBP, EP300, HDAC1 and HDAC2 were broadly distributed in both dental epithelium and mesenchyme.

Figure 5 Analysis of expression profiles for histone acetylation-related enzymes during tooth development.

(A) Histone acetylation marks and correlated acetyltransferases and histone deacetylases (B–E) Bubble plot analysis of select histone acetylation-related enzymes (corresponding to A) by scRNA-seq in the developing molar tooth germ at E12.5 (B), E14.5 (C), E16.5 (D), and P1 (E).

Figure 6 Localization patten of CREBBP, EP300, HDAC1 and HDAC2 during tooth development.

(A–D) Immunohistochemistry staining of CREBBP in developing molar tooth germ at E12.5 (A), E14.5 (B), E16.5 (C) and P1 (D). (E–H) Immunohistochemistry staining of EP300 in developing molar tooth germ at E12.5 (E), E14.5 (F), E16.5 (G) and P1 (H). (I–L) Immunohistochemistry staining of HDAC1 in developing molar tooth germ at E12.5 (I), E14.5 (J), E16.5 (K) and P1 (L). (M–P) Immunohistochemistry staining of HDAC2 in developing molar tooth germ at E12.5 (M), E14.5 (N), E16.5 (O) and P1 (P). Dashed lines outline the molar epithelium. Representative images are shown. Scale bar in (P) represents 50 µm.

Discussion

Intercellular signaling transduction plays an important role throughout the different stages of tooth development, which has already been shown to be influenced by both genetic and epigenetic regulation. Research on the epigenetic mechanisms in tooth development mainly focus on the regulation of DNA methylation and histone methylation modification (Zheng et al., 2014; Yuan et al., 2022). In addition to these mechanisms, histone acetylation represents another important epigenetic modification involved in the craniofacial bone and cartilage development. In this study, histone acetylation modification in the mandibular first molars at different developmental stages is systematically observed through immunofluorescence staining, of which the results confirm the dynamic regulation of histone acetylation during tooth formation.

High levels of histones H3 and H4 acetylation are associated with an open chromatin structure, facilitating active gene transcription. From the bud stage to the early bell stage, H3ac distribution gradually increased in both the dental epithelium and mesenchyme, while H4ac distribution exhibited higher than H3ac starting from bud stage and remained at relatively high levels throughout. It is known that varieties of signaling pathways including Shh, Wnt, Fgf are involved in early tooth development. Whether H4ac and/or H3ac interact with these signaling pathways in regulating early tooth development still requires further investigations. At late bell stage, H3ac and H4ac were predominantly expressed in the IEE layer and ameloblasts, while H4ac was also present in the stellate reticulum and OEE cells, suggesting that H3ac and H4ac function primarily in dental epithelial cells rather than in mesenchymal cells at late bell stage. Further researches are necessary to explore the specific mechanism for the H3ac and H4ac in regulating enamel formation.

Differential acetylation histones at various sites exerts distinct effects on chromatin structure and DNA transcription. Acetylation of histone H3 primarily occurs at Lys9, Lys14, Lys18, and Lys27. Specifically, acetylation at Lys9 is primarily involved in nucleosome positioning, while acetylation at other sites, such as Lys14, Lys18, and Lys27, is closely associated with transcription (Kuo et al., 1996; Roth & Allis, 1996; Zhang et al., 1998). Similarly, acetylation of histone H4 is most common at Lys5, Lys8, Lys12, and Lys16. Acetylation at Lys5 and Lys12 has been associated with nucleosomes localization on newly synthesized chromatin during S phase, while acetylation at Lys8 and Lys16 is predominantly present in chromatin with transcriptional activity (Turner, 1993; Sobel et al., 1995; Roth & Allis, 1996; Wittschieben et al., 2000). Among these modifications, H3K9ac, H3K27ac, H4K5ac, and H4K16ac have been focus of extensive research. In our study, the distribution patterns of H3K9ac, H3K27ac, H4K5ac, and H4K16ac were different to some extent throughout tooth development, suggesting a potential functional difference between these histone acetylation marks in mouse tooth development. Particularly, H3K27ac showed elevated distribution in dental papilla cells at E14.5 and E16.5, as well as in dental epithelial cells at E16.5. H4K16ac was prominently expressed in dental epithelium and dental papilla at E14.5 and E16.5. Given that dental papilla may play a role in the development of tooth shape (Kollar & Baird, 1969), it suggests that H3K27ac and H4K16ac may be key regulators of tooth morphology determination. Starting from E16.5, H3K9ac and H4K5ac were widely expressed in the dental epithelium, whereas H3K27ac is more concentrated in the enamel epithelial layer and ameloblasts. This distribution pattern suggests a pivotal role of H3K27ac in the differentiation of dental epithelium into enamel producing cells. In the mesenchyme, the elevated localization of H3K9ac and H3K27ac were observed during odontoblast differentiation (Tao et al., 2019a). Since the chromatin accessibility of markers like Dmp1 and Dspp increased before their transcriptional activation (Zhang et al., 2021), it is likely that H3K9ac and H3K27ac may function in odontogenic differentiation by modulating chromatin accessibility. Given the known association of H3K9ac and H3K27ac with active enhancers (Calo & Wysocka, 2013), further studies on active enhancers in regulating odontogenic differentiation is required. Previous studies have also unveiled that H4K5ac and H4K16ac play critical roles in early organ development, such as neural tube, placental, skin (Li et al., 2019; Wang et al., 2023; Bi et al., 2024). In the present study, we identified a widespread distribution pattern of H4K5ac and H4K16ac throughout the tooth development, while the function of H4K5ac and H4K16ac remains largely unexplored.

Histone acetylation levels are regulated by the balanced actions of HATs and HDACs. HATs include the GCN5 related N-acetyltransferase family, MYST family, EP300/CREB-binding protein (CREBBP), TAF250, Steroid receptor coactivator family (Shvedunova & Akhtar, 2022; He, Li & Li, 2023). HDACs are classified into four classes. Class I includes HDAC1, 2, 3, and 8; Class II includes HDAC4, 5, 6, 7, 9, and 10; and Class IV contains only HDAC11. Classes I, II, and IV are all Zn2+ dependent (Bondarev et al., 2021), while Class III, also known as Sirtuins, requires NAD+ for activity (Brancolini, Gagliano & Minisini, 2022). In the current study, by utilizing published scRNA-seq data, we identified that CREBBP, EP300, HDAC1 and HDAC2 exhibited higher expression levels than other HATs and HDACs during tooth development. Subsequent immunohistochemical staining confirmed the widespread distribution of CREBBP, EP300, HDAC1, and HDAC2 in both the dental epithelium and mesenchyme, although their distribution patterns did not fully align with the scRNA-seq data. The differences could be due to the use of independent experimental samples. Acetyltransferase EP300/CREBBP has been shown to express throughout the entire process of tooth development. In ameloblast cell lines, fluoride can enhance the activity of EP300, giving rise to histone acetylation at p53 binding sites, thus inhibiting the growth of ameloblasts (Deng et al., 2020). However, most research on histone acetylation has focused on the odontogenic differentiation of dental pulp cells (DPCs). During this process, EP300 is recruited to the promoter regions of osteocalcin and DSPP, which upregulates the acetylation of their histone H3K9, thereby promoting the upregulation of dentin related genes DMP1, DSPP, DSP, osteopontin, and osteocalcin (Wang et al., 2014). When EP300 is knocked out in DPCs, their proliferation and odontogenic differentiation are inhibited (Liu et al., 2015).

A preivous study showed that HDAC2 was expressed in a subset of DPCs, with strong expression in most mature odontoblasts. Inhibiting HDAC2 significantly upregulates the expression of osteopontin and BSP, while downregulating the expression of osteocalcin (Paino et al., 2014). In addition, HDAC3 and p300 are recruited by KLF4 to bind the promoter regions of DMP and SP7 to regulate transcription activation (Tao et al., 2019b). During the process of odontogenic differentiation, HDAC3 decreases while EP300 increases (Tao et al., 2020). Overexpression of HDAC5 leads to reduction in odontogenic/osteogenic related proteins and diminished formation of mineralized nodules, whereas inhibition of HDAC6, SIRT1, and SIRT6 (Kim et al., 2012; Sun et al., 2014; Wang et al., 2018b) suppresses odontogenic differentiation of DPCs. Histone deacetylase inhibitors HDACi, including SAHA and Trichostatin A, can induce odontogenic differentiation of DPSCs, promoting formation of dentin and dentin related matrix (Jin et al., 2013). Interestingly, SIRT6 knockdown mice does not exhibit a phenotype of abnormal tooth development, while overexpression of SIRT1 induces predentin maturation and dentin formation. This is further supported by the observation that the overexpression of Sirt1 in MSCs rescues or partially rescues dentin formation defects caused by Bmi1 deficiency (Wang et al., 2018a). Moreover, a recent study observed strong expression of HDAC4, HDAC5 and HDAC6 in postnatal rat DPCs and odontoblasts, with HDAC4 expression weakened in adult rat teeth (Yamauchi, Shimizu & Duncan, 2024). Conditional knockout of HDAC4 in dental follicle cells leads to significantly shortened tooth roots in mice, suggesting that HDAC4 has a regulatory role in tooth root development (Yamauchi et al., 2020). While previous studies have demonstrated the essential roles of HATs and HDACs in tooth development, it remains unclear how these enzymes specifically regulate histone acetylation modifications in this process, as individual HATs and HDACs can act on multiple histone acetylation modification sites. Knockout mice strains for these enzymes would be necessary to be established and characterized. Understanding this complex network of regulatory mechanisms will be the future challenge in histone acetylation research in tooth development.

Conclusion

Take together, these experiments provide the novel spatiotemporal pattern of histone acetylation marks in mouse molar development, suggesting histone acetylation may play an important role during tooth development.

Supplemental Information

Supplemental Information 1 ARRIVE 2.0 Checklist.

Supplemental Information 2 Quantification of H3ac, H3K9ac and H3K27ac expression at different timepoint of tooth development.

(A) Quantification of H3ac expression in dental epithelium and mesenchyme/ dental papilla at E13.5, E14.5, E16.5, E18.5 and P3. (B) Quantification of H3K9ac expression in dental epithelium and mesenchyme/ dental papilla at E13.5, E14.5, E16.5, E18.5 and P3. (C) Quantification of H3K27ac expression in dental epithelium and mesenchyme/ dental papilla E13.5, E14.5, E16.5, E18.5 and P3.

Epi, epithelium; Mes/DP, mesenchyme/ dental papilla; IEE, inner enamel epithelium.

Supplemental Information 3 Quantification of H4ac, H4K5ac and H4K16ac expression at different timepoint of tooth development.

(A) Quantification of H4ac expression in dental epithelium and mesenchyme/ dental papilla at E13.5, E14.5, E16.5, E18.5 and P3. (B) Quantification of H4K5ac expression in dental epithelium and mesenchyme/ dental papilla at E13.5, E14.5, E16.5, E18.5 and P3. (C) Quantification of H4K16ac expression in dental epithelium and mesenchyme/ dental papilla E13.5, E14.5, E16.5, E18.5 and P3.

Epi, epithelium; Mes/DP, mesenchyme/ dental papilla; IEE, inner enamel epithelium.

Supplemental Information 4 Quantification of CREBBP, EP300, HDAC1 and HDAC2 expression at different timepoint of tooth development.

(A) Quantification of CREBBP expression in dental epithelium and mesenchyme at E12.5, E14.5, E16.5 and P1. (B) Quantification of EP300 expression in dental epithelium and mesenchyme at E12.5, E14.5, E16.5 and P1. (C) Quantification of HDAC1 expression in dental epithelium and mesenchyme at E12.5, E14.5, E16.5 and P1. (D) Quantification of HDAC2 expression in dental epithelium and mesenchyme at E12.5, E14.5, E16.5 and P1.

Epi, epithelium; Mes, mesenchyme.

Supplemental Information 5 Raw data of Figures S1, S2, S3.

Each data point indicates the determine the average immunofluorescence pixel intensity of epithelium and mesenchyme/dental papilla for each antibody at different time point.

We thank Qinglu Tian and Siying Li for helpful discussion and assistance with the mouse colony.

Additional Information and Declarations

Competing Interests

The authors declare that they have no competing interests.

Author Contributions

Wen Du conceived and designed the experiments, performed the experiments, prepared figures and/or tables, authored or reviewed drafts of the article, and approved the final draft.

Wanyi Luo performed the experiments, analyzed the data, prepared figures and/or tables, and approved the final draft.

Liwei Zheng analyzed the data, authored or reviewed drafts of the article, and approved the final draft.

Xuedong Zhou conceived and designed the experiments, authored or reviewed drafts of the article, and approved the final draft.

Wei Du conceived and designed the experiments, analyzed the data, prepared figures and/or tables, authored or reviewed drafts of the article, and approved the final draft.

Animal Ethics

The following information was supplied relating to ethical approvals (i.e., approving body and any reference numbers):

Ethics Review Committee of the West China School of Stomatology, Sichuan University (WCHSIRB-D-2022-155).

Data Availability

The following information was supplied regarding data availability:

The raw measurements are available in the Supplemental File.

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
