# Peer review of "Temporal and spatial distribution of histone acetylation in mouse molar development"

_PeerJ, doi:10.7717/peerj.19215_

## Round 0.1 · original submission · Major Revisions

Thank you very much for your submission. All authors agree that the paper is interesting and has the potential to provide very important data. I agree with the assessment of Reviewer 2 that quantification of the current qualitative data would significantly strengthen the manuscript. Please do your best to address their concerns. I anticipate your resubmission would require re-review by Reviewer 2 if they are willing. I look forward to receiving your revised manuscript.

Reviewer 1 ·

Basic reporting

This is a very interesting and important study showing the histone methylation during mouse molar development. The methodology is adequate and the authors should mention other studies (less comprehensive ) on the histone analysis in tooth formation (Yamauchi Y, Shimizu E, Duncan HF. Dynamic Alterations in Acetylation and Modulation of Histone Deacetylase Expression Evident in the Dentine-Pulp Complex during Dentinogenesis. Int J Mol Sci. 2024 Jun 14;25(12):6569. doi: 10.3390/ijms25126569. PMID: 38928274; PMCID: PMC11203584., Yuan H, Suzuki S, Terui H, Hirata-Tsuchiya S, Nemoto E, Yamasaki K, Saito M, Shiba H, Aiba S, Yamada S. Loss of IκBζ Drives Dentin Formation via Altered H3K4me3 Status. J Dent Res. 2022 Jul;101(8):951-961. doi: 10.1177/00220345221075968. Epub 2022 Feb 22. PMID: 35193410.).

The English language is adequate.

The figures are beautiful and clearly show the expression of the acetylated histones on the tissues analyses

This referee believes that the manuscript deserves publication on Peer J after these small modifications suggested.

Experimental design

The methods used are adequate and the results ares clearly demosntrated by excelent quality figures

Validity of the findings

The results are new and will add to the knowledge on the field. The conclusions are backed by the results.

Additional comments

Sugges acceptance efter minor revision (include 2 references). No need for further review.

Reviewer 2 ·

Basic reporting

The manuscript by Wen Du et al., presents histone acetyllysine-antibody immunofluorescence at multiple stages in mouse tooth development. Specifically, the authors compared immunofluorescence of -pan-H3ac, -pan-H4ac and individual lysine acetyl antibodies to each other, between different parts of the developing tooth, and at different time points. The authors conclude that histone acetylation likely plays an important role in tooth development, which may be directed by specific actyllysine modifications under spatial and temporal regulation.

Strengths:
The difficulty in acquiring the relevant tissue and performing paraffin sectioning is appreciated, and the resulting immunofluorescent images are beautiful. The discussion is a comprehensive summary of the intersection of odontogenic differentiation and epigenetics.

Weaknesses:
There are two major technical concerns that I believe are necessary for acceptance, and one major epistemological concern that may or may not be able to be addressed, but I would like to see the authors attempt to.

Technical concern #1: The data described in the text is based on the intensity of immunofluorescent signal per antibody, cell type and developmental stage. The authors present these data as being more or less “expressed”, but I could find no evidence of quantification. The representative images are nice, but they are not intuitive regarding cell type or morphology for a non-specialist, and more importantly, the reader cannot properly judge whether the fluorescent signal is truly more intense or not. This requires quantification of fluorescent signal and appropriate presentation of those measurements in graphs.

There is also no mention of how the signal was normalized, which is essential to interpret the data (i.e., weak fluorescent signal can be enhanced computationally to make it seem strong). I am not suggesting that there is any foul play by the authors, I’m just making the point that these data need to be normalized systematically between experiments and replicates, and that their method of normalization needs to be documented. Finally, proper interpretation of the data requires statistical comparisons between cells, times and antibodies – which brings me to my second point:

Technical concern #2: The authors state in their methods section that “each experiment was performed at least three times with different embryos”. However, that is the sole reference to replication in the entire manuscript. The authors should quantify the intensity of each experiment and compare the means of replicates, rather than just state a given modification was more or less expressed based on a representative figure. Without presenting replication of the data and statistical comparisons, the reader can only go by the word of the authors that the main text figures are representative. The replicates that are not part of the main figures should also be put into supplementary files so that an interested reader can assess the repeatability of the experiment.

Epistemological concern: Ostensibly, the goal of this study was to see whether histone acetylation plays a role in tooth development. How does this study show that? How does showing different patterns of histone acetylation confirm of reject a hypothesis that acetylation has a role in tooth development? At the resolution of cells visualized by microscopy, these data do not meaningfully inform the abundance of these marks at genes involved in tooth differentiation – which is the relevant information needed to make correlational inferences.

I am a proponent of building experimental systems slowly and methodically, so if the argument is that future work will use this technique and data as part of an experiment to identify if and how histone acetylation influences tooth development, then I would be more enthusiastic about this manuscript. Alternatively, if the authors have a good rebuttal on why these data are useful, I would be happy to hear it. At a minimum, I would like to see the authors add to their Discussion, Conclusion or Introduction at least one “next step” that will benefit from the current manuscript and lead towards knowledge regarding histone acetylation and tooth development.


Minor comments

o I think the word “epigenetic” in the abstract on Line 23 should be changed to “chromatin” or “histone”, as strictly speaking there is little mechanistic evidence that acetylation is passed down through cell division, in contrast to the repressive histone modifications H3K9me2/3 and H3K27me3 (Margueron & Reinberg, 2010) and small RNAs.

o “are” in line 25 of Abstract: the role of histone acetylation in tooth development are poorly understood” should be “is”

o Line 198, remove the ‘s’ from ‘genetics and epigenetics regulation’

o Line 48 and 49 in the Introduction: I’m not sure if histone modification has been “most investigated” of the epigenetic mechanisms, or if its even important to make that claim here.

o In the Introduction the authors claim that relatively little research on histone acetylation has been conducted regarding tooth development. Yet in the Discussion lines 269-288 the authors cite several important papers showing there is such a connection. I recommend referencing at least some of these in the Introduction as well.

Experimental design

all comments made in "basic reporting" section

Validity of the findings

all comments made in "basic reporting" section

Additional comments

all comments made in "basic reporting" section

Reviewer 3 ·

Basic reporting

1) In this paper, the authors used “expression” a lot, but “modification” would be appropriate.

Experimental design

1) The authors showed expression variation of enzymes related to histone acetylation by reanalysis of scRNA-seq data. Please provide immunobiological data for localization of several enzymes with high expression levels. The data would further support the reanalysis results.

Validity of the findings

no comment

---

## Round 0.2 · Minor Revisions

We received one new review of your revised manuscript. Many of the previous concerns are addressed. However, there are two concerns that I would like to give you the opportunity to address or respond to.

1) Could you please provide the rationale for not including the histological images demonstrating variability as supplementary figures? They seem to be prepared already.

2) I am somewhat sympathetic to the precise use of the term "expression" for referring to gene transcription. Reviewer 3 previously mentioned "modification" as a description. Would it also be appropriate to describe your IHC experiments as identifying the "localization" of histone modification as opposed to "expression"?

I look forward to your quick revisions or responses to these straightforward concerns.

Reviewer 2 ·

Basic reporting

The addition of Supplementary Figures 1 & 2 address most of my concern regarding quantification and reproducibility. However, the authors have elected not to add the fluorescent images themselves into new supplementary figures. I think they should, but leave that up to the editor. I am also not totally convinced by their answer to my third major comment (see 'Epistemological concern'), but I also see that as being up to the editor to decide.

One final small comment; as reviewer #3 pointed out, they often refer to histone modification abundance as "expression" including in the title. It would be more appropriate to use the words 'abundant' or 'modified' as 'expression' is much more commonly used to refer to transcription /mRNA.

Experimental design

no comment

Validity of the findings

no comment

Additional comments

no comment

---

## Round 0.3 · accepted · Accept

Thank you for your quick revisions to the manuscript. I now recommend that it be accepted. Thank you for your patience and your careful response to the reviewers' suggestions.